# The ExPAND Study: A Prospective Association Study into Endometriosis-Associated Pain, Neurosteroid Synthesis, and TRPM3

**DOI:** 10.3390/biom15101352

**Published:** 2025-09-23

**Authors:** Eleonora Persoons, Celine Bafort, Pilar Van Mechelen, Martina Ciprietti, Katrien Luyten, Melissa Benoit, Arne Vanhie, Thomas Voets, Carla Tomassetti, Joris Vriens

**Affiliations:** 1Implantation, Placentation, Pregnancy and Endometriosis (POPPYe) Research Group, Department of Development and Regeneration, KU Leuven, 3000 Leuven, Belgium; eleonora.persoons@kuleuven.be (E.P.); celine.bafort@uzleuven.be (C.B.); pilar.vanmechelen@kuleuven.be (P.V.M.); martina.ciprietti@kuleuven.be (M.C.); katrien.luyten@kuleuven.be (K.L.); arne.vanhie@uzleuven.be (A.V.); carla.tomassetti@uzleuven.be (C.T.); 2Laboratory of Ion Channel Research (LICR), VIB-KU Leuven Center for Brain & Disease Research, Belgium & Department of Cellular and Molecular Medicine, KU Leuven, 3000 Leuven, Belgium; melissa.benoit@kuleuven.be (M.B.); thomas.voets@kuleuven.be (T.V.); 3Leuven University Fertility Center, University Hospitals Leuven, Herestraat 49, 3000 Leuven, Belgium

**Keywords:** DHEAS, PS, endometriosis, pain, steroidogenesis, TRPM3

## Abstract

Endometriosis-associated pain has debilitating effects on the quality of life of patients. Despite its high prevalence in reproductive-aged women, the pathophysiology is still unknown, impeding the development of targeted treatment approaches. The prospective ExPAND study proposes the neurosteroids pregnenolone sulphate (PS) and dehydroepiandrosterone sulphate (DHEAS) as potential contributors to endometriosis-associated pain, due to their agonistic action at the pain-related ion channel TRPM3. To this end, endometrium, deep endometriosis lesions, and peritoneal fluid were prospectively collected in four demarcated patient groups, which were characterised based on their pain symptoms, as scored via the WERF-EPHect questionnaire, i.e., (1) control (*n* = 44), (2) endometriosis patients with no pain symptoms (*n* = 24), (3) with only severe dysmenorrhea (*n* = 54), or (4) with both severe dysmenorrhea and non-cyclic pelvic pain (*n* = 78). Tissue mRNA expression levels of steroidogenic enzymes were investigated and showed significantly increased levels of CYP17A1 in the endometrium of patients with severe pain symptoms compared to control tissue. In addition, liquid chromatography with tandem mass spectrometry (LC-MS/MS) was performed to investigate neurosteroid concentrations in the peritoneal fluid. Both neurosteroids PS and DHEAS were present in the peritoneal fluid at concentrations that are known to stimulate TRPM3 activity in vitro. Finally, using microfluorimetric Ca^2+^ imaging, we demonstrate that both DHEAS and PS stimulate human stem-cell-derived sensory neurons in a TRPM3-dependent manner. Taken together, these data indicate a potential contribution of steroidogenesis and TRPM3 in endometriosis-associated pain.

## 1. Introduction

Endometriosis is a common (incidence: 10%) gynaecological disease and is characterised by the growth of ectopic hormone-responsive endometrial-like tissue in an inflammatory environment [1]. There are three different types of lesions, i.e., superficial peritoneal, ovarian, and deep, which are all associated with many pain symptoms, like dysmenorrhea and non-cyclic pelvic pain, impacting the patient’s quality of life. In general, disease staging systems (e.g., rAFS or #Enzian) do not fully correlate with pain symptoms [2,3]. In addition, endometriosis-associated pain is thought to have nociceptive, inflammatory, and central mechanisms, giving the pain a complex character [4,5]. Investigating each component separately might help to obtain better therapeutic interventions [6]. Endometriosis lesion proliferation is oestrogen-driven, defining it as steroid-dependent [7]. Indeed, genomic studies advocate for a role of sex steroids in the pathophysiology [8]. Oestrogen provisioning occurs through three main mechanisms: (i) systemic ovarian production, (ii) local formation due to an altered steroidogenic pathway, e.g., StAR, CYP19A1, or HSD17β1 upregulation [9], and (iii) conversion of adrenal precursor steroids by the lesion [10,11]. In this regard, the production, activation, and/or metabolism of dehydroepiandrosterone (DHEA) is of interest. Primary endometriosis cultures can convert DHEA into androstenedione due to increased 3β-HSD expression [12]. Moreover, pregnenolone and DHEA are increased in endometriosis tissue compared to serum concentrations [12].

Aside from proliferation, oestrogens also contribute to neurogenesis of Aδ- and C-nociceptive neurons [6,13,14,15]. These serve as mediators in the transmission of painful stimuli and are equipped with a repertoire of specific ion channels which serve as pain transducers. In endometriosis, the expression pattern of these nociceptors is still under investigation. Here, members of the transient receptor potential (TRP) ion channel superfamily have been suggested, as some have already been identified to be involved in peripheral pain sensing, e.g., TRPV1 and TRPA1 [16,17]. Indeed, the density of TRPV1-positive nerve fibres in ovarian endometrioma is associated with pelvic pain severity [18] and TRPV1 mRNA is significantly upregulated in the peritoneum of women with endometriosis [19]. Unfortunately, current antagonists of TRPV1 have failed so far in clinical trials for the treatment of peripheral chronic pain [20]. In addition, an elevation of TRPA1 mRNA expression levels was reported in peritoneal tissue of endometriosis, but not within the endometriosis lesions [19].

In the last decade, TRPM3 has emerged as a transducer for peripheral inflammatory pain. TRPM3 is a calcium-permeable cation channel, expressed in around 60% of the human dorsal root ganglia (DRG) and involved in inflammatory hyperalgesia [21,22,23]. Recent research has shown that TRPM3 is upregulated in DRG neurons that innervate inflamed tissue [21,24]. Interestingly, the endogenous neurosteroids pregnenolone sulphate (PS), DHEA, and DHEA-sulphate (DHEAS), which are presumably involved in endometriosis’ pathophysiology, are also direct agonists of TRPM3 [25,26,27,28]. Given the inflammatory and steroid-dependent character of endometriosis, we hypothesised that these neurosteroids might contribute to endometriosis-associated pain through activation of TRPM3 in sensory nerve endings that innervate the lesions. In particular, we questioned whether the severity of pain symptoms in endometriosis patients is associated with the production of DHEAS or PS. To this end, the monocentric prospective Endometriosis, Pain and Neurosteroids (ExPAND) study was set out aiming to shed light on parts of the nociceptive and inflammatory components of endometriosis-associated pain.

## 2. Materials and Methods

### 2.1. Ethical Approval

The use of human biological residual tissues was approved by the UZ/KU Leuven Ethical Committee (S62818). Additionally, this study was conducted in accordance with the Declaration of Helsinki, Good Clinical Practice guidelines, and all relevant regulatory requirements. Written informed consent was obtained before collecting endometrial biopsies, endometriosis lesions, or peritoneal fluid.

### 2.2. Eligibility Criteria and Patient Demographics

Between January 2020 and February 2023, patients who underwent laparoscopic surgery for endometriosis-associated pain symptoms, fertility issues, and/or other gynaecologic treatments at the UZ Leuven, Belgium were screened for eligibility after signing the informed consent form. Patients were included based on pain scores and the presence or absence of (deep) endometriosis. The participants in this study were women of reproductive age who underwent laparoscopic surgery for a gynaecologic disorder or treatment at the Department of Gynaecology and Obstetrics in University Hospital Leuven in Belgium. The enrolled patients were divided into four different groups based on findings during laparoscopic surgery and their pain symptoms. The control group (Group 1) consisted of women without pain symptoms, i.e., women who consulted their physician for other reasons, e.g., infertility issues or sterilisation, and where no signs of endometriosis were found during laparoscopic surgery. Women diagnosed with deep endometriosis during laparoscopic surgery were included in either Group 2, 3, or 4, based on their symptoms. Patients consulting the clinic for fertility issues without important endometriosis-associated pain symptoms were assigned to Group 2. Women who report severe dysmenorrhea (DYSM) were included in Group 3, while Group 4 consisted of women with both DYSM and non-cyclic pelvic pain (NCPP). DYSM and NCPP were defined by patient-reported pain outcomes (PROMs) using an 11-point numeric rating scale (NRS), i.e., 0–10, at time of the initial clinical consultation. These NRSs were provided through the World Endometriosis Research Foundation—Endometriosis Phenome and Biobanking Harmonisation Project (WERF-EPHect) endometriosis patient questionnaire—standard (EPQ-S) [29]. Here, question C13 rates the severity of pain during menstruation when it was at its worst. A score of 7 or higher was classified as DYSM. This high inclusion threshold is largely based on the study of Kural et al., which showed that 84.4% will report pain during their menstruation. Using a NRS, they could demarcate mild (NRS 1–3) and moderate (NRS 4–7) from severe (NRS 8–10) DYSM, showing that only 34.2% of woman experienced severe menstrual pain [30]. Given the chronic character of non-cyclic pelvic pain, the inclusion threshold was set lower, i.e., NRS ≥ 5 on question C39 of the WERF-EPHect questionnaire. In Group 3, the NRS score for dysmenorrhea was 8.5 ± 1.0, whereas the NRS score for non-cyclic pelvic pain was low (0.7 ± 1.2). This is also illustrated by the reported use of pain medication for each indication (Table 1). In contrast, the NRS scores in Group 4 are high for both dysmenorrhea (8.8 ± 0.8) and non-cyclic pelvic pain (8.3 ± 1.4) (Table 1). Interestingly, the NCPP of Group 4 is on average at the same pain levels of severe DYSM. Regarding the nature of the endometriosis lesions, both the rAFS and #Enzian score were considered and were found to be comparable between groups. Some commonly used therapies could also influence circulating levels of DHEA and DHEAS. For example, methylphenidate could increase DHEAS levels [31], while dexamethasone could decrease DHEAS serum levels [32]. However, none of the included patients disclosed the use of such drugs. In addition, polycystic ovarian syndrome (PCOS) is presented by increased androgens as well as DHEAS. However, patients with reported PCOS in our study cohort (3.5%) did not show increased peritoneal DHEAS levels and were therefore not excluded.

In total, 200 women participated in the study. The sample size per analysis is summarised in Appendix A.

### 2.3. Analysis of Single-Cell Datasets

Single-cell RNA sequencing data were obtained from the paper of Fonseca et al. and analysed [33]. Six datasets were utilised, including eutopic endometrium from two healthy control individuals (GSM6574512, GSM6574543), eutopic endometrium from two endometriosis patients (GSM6574528, GSM6574534), and paired endometriosis tissue samples from the same patients (GSM6574530: deep endometriosis lesion in the uterosacral ligament; GSM6574532: deep endometriosis lesions in the pelvic side wall). These endometriosis datasets were the only ones from the Fonseca et al. [33]. study where both EB and deep EL were collected from the same patient, thereby reflecting our study setup [33]. Data preprocessing and analysis were conducted using the Seurat R package (v4.1). Raw gene expression matrices were imported using the Read10X function, and Seurat objects were created. Data normalisation and identification of highly variable features were performed. The datasets were merged into two Seurat objects using the merge function, with unique identifiers assigned to distinguish cells from different patients. Batch effects were corrected using Harmony integration [34], grouping cells by sample identity. Principal component analysis was applied to scaled data, followed by construction of a neighbourhood graph using Harmony-reduced dimensions. Cell clusters were identified at a resolution of 0.5, and dimensionality reduction was visualised via UMAP. Cluster identities were refined based on known marker gene expression patterns and markers from the original paper [33,35]. Specific clusters, including fibroblasts, epithelial cells, stromal cells, immune cells, endothelial cells, perivascular cells, and red blood cells, were annotated iteratively using conditional statements and visualised with UMAP. When marker-based annotation was unclear, differential gene expression was analysed using the FindMarkers function, with parameters min.pct = 0.25 and logfc.threshold = 0.25 (Appendix A).

### 2.4. Endometrial and Endometriosis Samples

#### 2.4.1. Collection

During laparoscopic surgery endometrial biopsies were obtained using a sterile Novak curette. A small piece of freshly removed endometrial biopsy was used for diagnostic ends. The remnants were kept on RNALater (Qiagen, Venlo, The Netherlands) for 24 h to preserve RNA and stored at −80 °C until real-time quantitative polymerase chain reaction (RT-qPCR) analysis was carried out. In endometriosis patients, deep infiltrating endometriosis nodules were excised during laparoscopic surgery using a CO_2_ laser. One piece of such nodules was used for diagnostic ends to confirm the endometriosis diagnosis. Indeed, immune histochemistry showed that all nodules in this study were CD10^+^ and cytokeratin^+^ and thereby diagnosed as endometriosis by pathologists at the University Hospitals of Leuven, Belgium. Part of the nodule was kept on RNALater for 24 h and stored at −80 °C until further RT-qPCR analysis. The remnants were placed on 4% PFA to fixate for 24 h, whereafter they were stored on 70% ethanol until further IHC analysis.

#### 2.4.2. RNA Isolation

RNA was isolated from frozen whole endometrial biopsies and endometriosis lesions using previously established protocols from our research group [36]. These were performed in batches every trimester to limit long-term storage which could impact expression patterns. Briefly, a power homogeniser (Polytron, Montreal, QC, Canada) was used to homogenise the tissue. Next, total RNA was extracted using the TriPure Nanodrop Method (Roche, Mannheim, Germany) and the RNA quality was assessed using an Experion RNA StdSens Analysing kit (Isogen Life Science, Temse, Belgium). Only samples in which the RNA integrity score ≥ 7 were included. Isolated RNA was kept at −80 °C for a maximum of 1 year, whereafter it was processed into cDNA and RT-qPCR was performed.

#### 2.4.3. cDNA Preparation and RT-qPCR Experiments

One microgram of the extracted RNA was used for cDNA synthesis using the High-Capacity cDNA Reverse Transcription Kit (Life Technologies Europe B.V., Ghent, Belgium). RT-qPCR experiments were carried out using the previously established protocols from our research group [36]. In short, samples of triplicate cDNA, diluted 2.5-fold, were analysed using the StepOne PCR system (Applied Biosystems, Life Technology, Carlsbad, CA, USA) with specific Taqman gene expression assays (Appendix A). For each gene, the reaction mixture included 2 µL cDNA, 5 µL Mastermix (Life Technology), 2.5 µL diethylpyrocarbonate-treated RNase-free water, and 0.5 µL of the corresponding TaqMan primer. GAPDH, HPRT1, and TBP were used as endogenous controls, selected based on GeNORM algorithm analysis of pilot samples. The thermal cyclin protocol consisted of an initial holding stage at 95 °C for 20 min, followed by 40 replication cycles at 60 °C for 20 min. Genes with cycle threshold (C_T_) values equal to or greater than 40 were considered non-detectable. Data were shown as 2^(−ΔCT)^ (mean ± SD) in which ΔC_T_ = C_T enzyme_ − C_T geometric mean of endogenous controls_. Statistical tests were performed on the ΔC_T_ values and are further indicated in the figure legends.

### 2.5. Peritoneal Fluid Samples

#### 2.5.1. Collection

During laparoscopic surgery peritoneal fluid was obtained using either suction devices or a laparoscopic blunt needle and manual aspiration using a syringe. Note that peritoneal fluid was collected before any other surgical intervention (e.g., hysteroscopy) to prevent contamination with blood or saline washes. After collection, the sample was processed by centrifugation at 1400 rpm for 10 min at 4 °C. The supernatant was stored at −80 °C until further analysis could be carried out.

#### 2.5.2. Quantitative Bioanalysis by LC-MS/MS

These experiments were performed by ASAS Labor GmbH (Köln, Germany). In brief, for each sample, 100 µL was fortified with 400 µL diluted internal standard solution (IS). After mixing, the samples were centrifuged for 3 min at 12,000 rpm. Of the supernatant, 10 µL was injected into the reversed phase HPLC system with MS/MS detection, using an ESI interface for ionisation in negative mode followed by MS/MS. The compounds were quantified by internal standardisation taking the peak area ratio of each of the analytes and of the IS as a response parameter. The HPLC consisted of an Agilent G1312A 1200 binary pump, a Agilent G1322A degasser, and an Shimadzu SIL-20AC autosampler. The chromatographic column was a Phenomenex Synergi Fusion-RP 80 A, 4 µm particle size, 50 mm (length) × 2.1 mm (inner diameter). Mobile phase A was 10 mM ammonia acetate in H_2_O and mobile phase B was methanol. The mass spectrometer used was a SCIEX API 4000 with a TurboIon Spray source, operated in negative mode. The obtained peak area ratios were fitted against the spiked concentrations of calibration standards, using a weighted quadratic fit. Calibration was performed from 44.4–2573 ng/mL for DHEAS and 4.83–251 ng/mL for PS, using five concentrations. For the analytes, the obtained peak area ratios were fitted against the spiked concentrations, using a weighted quadratic fit. The limit of quantification was 44.4 ng/mL for DHEAS and 4.83 ng/mL for PS. Quality control samples, prepared at 532 ng/mL DHEAS and 53.1 ng/mL PS, showed an accuracy and precision of 5.57% and 2.32% for DHEAS and −0.34% and 1.87% for PS.

### 2.6. Human Stem-Cell-Derived Neurons (hSCD Neurons)

#### 2.6.1. Cell Culture

hSCD neuronal cells were differentiated and cultured as previously described [23,37,38]. Confluent human embryonic stem cells were dissociated using Accutase and replated at a high density on Matrigel^®^ matrix. Once confluent, cells were induced into neuronal precursors by incubation with induction medium (Appendix A), which was replaced daily for 5 consecutive days. Thereafter, precursor specification was achieved by incubating the cells for 10–18 days with the induction medium, supplemented with CHIR99021, DAPT, SU5402, LDN193189, and SB-431542 (replacing the medium daily) (Appendix A) [37,38]. Next, cells were replated into Matrigel^®^ matrix-coated Nunc™ Glass Bottom Dishes (after Accutase incubation). Maturation of the neuronal cells was achieved by incubation with the respective medium for 7 days (replaced three times a week) (Appendix A).

#### 2.6.2. Microfluorimetric Calcium Imaging

Standard Ca^2+^ imaging was performed following established protocols from our research group [16,36]. Briefly, intracellular Ca^2+^ levels were visualised by incubating the cells with 2 µM Fura-2 acetoxymethyl ester for 30 min at 37 °C prior to the measurement. During imaging, cells were maintained in a modified Krebs’ solution containing (in mM) 138 NaCl, 2 MgCl_2_, 2 CaCl_2_, 5.4 KCl, 10 glucose, 10 HEPES. TRPM3 channel activity was assessed by applying of agonists (DHEAS (100 mM stock) and PS (10 mM stock)). Isosakuranetin (ISOSA, 10 mM stock) was used as an antagonist. At the end of each experiment, a depolarising 50 mM K^+^ solution was applied to identify excitable cells. Throughout the measurement, the Krebs’ solution was maintained at 37 °C using a Peltier element. Fluorescent signals were elicited by alternating excitation at 340 and 380 nm using a Lambda XL illuminator (Sutter Instrument, USA)and captured with an Orca Flash 4.0 camera mounted on a Nikon Eclipse Ti fluorescence microscope. Imaging acquisition and ROI detection were performed with the NIS-Elements AR 5.42.07 (Built 1828) software (Nikon). Absolute Ca^2+^ concentrations were calculated from the fluorescence ratio (F340/F380), corrected for background fluorescence, using the Grynkiewicz equation [39]. Cells were considered responders if the Ca^2+^ influx during agonist application exceeded 20 nM and the peak derivative of the Ca^2+^ trace surpassed three times the standard deviation of the baseline derivative. Ca^2+^ amplitudes were defined as the difference between peak and basal Ca^2+^ levels in responding cells. Only cells that responded to KCl at the end of the experiment were included in the analysis.

### 2.7. Histology and Analysis

Histology was performed on fixed endometriosis lesions to determine inflammation markers. A set of six 4 µm thick slices was made every 100 µm and the first slice of each set was stained with haematoxylin and eosin (H&E). Using this stain, peripheral boundaries (i.e., transition from healthy to disease tissue) were determined based on the absence of glandular structures. All sets in between these boundaries were stained in sequence for CD68, cytokeratin, Masson trichrome, and PGP9.5.

#### 2.7.1. Immunohistochemistry Stains

Slides were subjected to a series of deparaffinisation and rehydration with, respectively, toluene and 100% ethanol. Details regarding the stains are summarised in Appendix A. Positive signals were visualised using the chromogen 3,3-diaminobezidine (Sigma). Sections were counterstained in haematoxylin. Analysis was performed with QuPath software (v0.6.0). In brief, positive cells were detected (for CD68 stain: cell DAB OD max of 0.8; for PGP9.5: threshold detection using DAB) in a radius of 200 µm around any glandular structure (based on the cytokeratin stain within the set). Percentages of positive cells were averaged within one sample.

#### 2.7.2. Masson Trichrome Stain

Masson trichrome staining was used to determine collagen fibres within the endometriosis lesions. Slides were subjected to a series of deparaffinisation and rehydration with, respectively, toluene and 100% ethanol. In sequence, Weigert’s iron haematoxylin working solution was applied for 10 min, Biebrich scarlet–acid fuchsin solution for 15 min, then samples were differentiated with phosphomolybdic–phosphotungstic acid solution for 10 min, aniline blue solution 5 min, and 5 min in 1% acetic acid solution. Then, they were briefly rinsed in distilled water and differentiated in 1% acetic acid solution for 5 min. Finally, dehydration was performed with ethyl alcohol and clearing in Neo-Clear. Collagen was stained blue, nuclei black, and muscle/cytoplasm and keratin red. Analysis was performed with QuPath software. The percentage of collagen-rich area in a radius of 200 µm around any glandular structure (based on the cytokeratin stain within the set) was determined. The percentage of collagen^+^ area was averaged within one sample.

### 2.8. Data Analysis, Statistics, and Figure Representation

Power calculations were carried out for a one-sided significance level of alpha of 5%, showing that a minimum of 104 subjects enrolled in the study provides a power of 95% to detect a difference of 20% in peritoneal fluid concentrations in the four groups (using an ANOVA test). The effect size of 0.42 is based on an upper bound within-group standard deviation of 2.9 μM, as provided by a preliminary study with two groups. To ensure an optimal sample size, the number of patients was increased to 120 subjects, 30 per group. Data analysis was performed in Excel. Calcium imaging was analysed with an in-house IGOR protocol. Statistical tests were performed with GraphPad Prism 10. Data was assessed for the presence of outliers using the robust regression and outlier removal (ROUT) method, Q = 1% [40]. Only in the datasets EL_SULT1E1_, PF_DHEAS_, and PF_PS_ were limited numbers of outliers detected. Analysis with and without these values did not change the statistical outcomes. Depending on data distribution (D’Agostino–Pearson normality omnibus K2 test), comparisons between multiple groups were performed with either one-way ANOVA or a Kruskal–Wallis test, which were both corrected for multiple comparisons. When comparing two groups of non-normal distribution or small sample size, a Mann–Whitney test was used. For the analysis of RT-qPCR expression data, ΔC_T_ values were used. In the text, data is presented as mean ± standard deviation, followed by the p-value of the test, if statistically significant. ANCOVAs were performed to assess group differences while controlling for potential confounders. Age was included as a covariate in the model. No other confounders (e.g., treatment or medication) were included in this specific analysis. The ANCOVA was performed using R software. All figures were made with GraphPad Prism 10, representing as mean ± standard error of the mean, unless mentioned otherwise.

## 3. Results

### 3.1. RNA Expression Profile of Steroidogenic Enzymes in Eutopic Endometrium and Deep Endometriosis Lesions

To assess if there is expression of the key enzymes in the metabolisation of pregnenolone and DHEA (i.e., CYP11A1 and CYP17A1) or PS and DHEAS (i.e., SULT1E1, SULT2A1, and SULT2B1) (Figure 1, Appendix A) in endometriosis, we first consulted a publicly available single-cell RNA sequencing dataset by Fonseca et al. [33], containing two healthy control individuals and paired eutopic endometrium and endometriosis tissue (EL) from two endometriosis patients. Indeed, when comparing endometrium of controls and patients, expression of CYP11A1 was observed in patient endometrium (Appendix A). When comparing eutopic and ectopic tissue from endometriosis patients, more cell types express CYP11A1, hinting towards PS (and DHEAS) in endometriosis pathophysiology (Appendix A). However, it must be noted that the menstrual cycle of the patient in this study was not considered and may impact the expression of enzymes involved in intracrine metabolism.

Publicly available datasets often do not report any details regarding patient-reported pain outcomes. Therefore, we collected eutopic endometrial biopsies (EBs) and ectopic endometriosis lesions (ELs) in our ExPAND study population to investigate whether differences in expression pattern could be detected regarding pain symptoms. In the eutopic endometrium biopsies, CYP17A1 expression was significantly different between groups, with an important upregulation in endometrial biopsies of patients with severe dysmenorrhea (Groups 3 (ΔC_T_ 8.5 ± 1.6) and 4 (ΔC_T_ 8.6 ± 1.8), compared to control Group 1 (ΔC_T_ 10.5 ± 2.2) (one-way ANOVA with Dunnett’s correction, *p* = 0.0002, and *p* = 0.0002, respectively)) (Figure 2A). Given that age could be a confounding factor in this difference, simple linear regression was used to test whether the participant’s age significantly predicted CYP17A1 expression levels. Here, the overall regression was not statistically significant (R^2^ = 0.02074, F(1131) = 2.775, *p* = 0.0981, Y = 0,04928*X + 7595), excluding age as a confounding factor in these findings. An additional ANCOVA for age as a covariate confirmed the main findings, showing a significant effect of the group (p_GROUP_ < 0.0001) and no significant effect of age (p_AGE_ = 0.7601), suggesting that the observed group differences are independent of age. These differences were not observed in the endometriosis lesions (Figure 2B).

For CYP11A1, SULT1E1, and SULT2B1, no difference in mRNA levels was observed in the endometrium, the endometriosis lesion, nor in the EL/EB ratio of the four study populations (Appendix A). In addition, the mRNA expression of SULT2A1 was not detectable in the eutopic endometrium (C_T_ = 40). However, when comparing EB and EL expression in a paired analysis of a smaller cohort, the EL/EB ratio was significantly different between Group 2 and Group 3 (ΔC_T_ EL/EB 1.8 ± 0.22 vs. 1.457 ± 0.32, Kruskal–Wallis test with Dunn’s multiple comparisons test, *p* = 0.0432) for CYP17A1 (Appendix A).

### 3.2. PS and DHEAS Levels in Human Peritoneal Fluid of Control and Endometriosis Patients

As the combined enzymatic activity of CYP11A1 and SULT2B1 or CYP17A1 and SULT1E1 results in the production of PS and DHEAS, respectively, from cholesterol, the concentrations of these neurosteroids were determined in the peritoneal fluid of control and endometriosis patients, using liquid chromatography with tandem mass spectrometry (LC-MS/MS). Interestingly, in all groups, PS and DHEAS were detected in the peritoneal fluid, yielding an average PS concentration of 42.7 ± 25.8 nM and a higher average for DHEAS of 2.6 ± 1.5 µM. No intergroup differences were identified for both neurosteroids (Figure 3). The different therapeutic hormonal therapies that endometriosis patients receive could potentially affect the neurosteroid production. Therefore, a subanalysis was performed, however, no differences between the therapies were observed (two-way ANOVA, *p*_MEDICATION_ = 0.1261; *p*_GROUP_ = 0.7155).

Even though there are no differences in neurosteroid concentration between our groups, TRPM3 expression could still be upregulated in patients with more severe pain symptoms. In this light, it is important to verify whether, at these nano- and micromolar range concentrations, neuronal TRPM3 could also be activated. Regarding PS, Vriens et al. reported that this neurosteroid can activate TRPM3 at doses as low as 100 nM [16]. Given that our reported average dose of 42 nM is measured in a peritoneal fluid sample (i.e., concentration is diluted), it could be expected that local intratissue levels are up to 10-fold higher [12,41]. In addition, TRPM3 is a temperature-sensitive ion channel, meaning that there is a potentiating effect of the channel activity at body core temperature. To investigate whether DHEAS also has the potential to activate endogenous TRPM3 at micromolar concentrations, embryonic H9 stem cells were differentiated into human neuronal cells [23]. Next, using a Fura-2-based imaging technique, TRPM3 functionality upon stimulation of DHEAS was investigated, at a previously reported dose of 100 µM [27]. Here, the application of 100 µM DHEAS at 37 °C induced a robust increase in intracellular calcium in PS-responding cells (Figure 4A). Note that the percentage of PS responders was in line with earlier reports, i.e., 77% [23]. The specificity of the DHEAS response was confirmed using two repetitive applications of DHEAS, with co-application of the TRPM3 antagonist isosakuranetin (ISOSA, 5 µM) during the first stimulus (Figure 4B). The application of 100 µM DHEAS results in an increase of intracellular calcium (144 ± 73 nM) in 38 ± 12% of PS-responding cells (Figure 4C; Figure 4D). In the presence of isosakuranetin, the amplitude of the calcium influx as well as the fraction of DHEAS-induced calcium responders was significantly reduced in hSCDS neuronal cells (28 ± 2 nM, Mann–Whitney, *p* = 0.002; 10 ± 2%, Mann–Whitney, *p* = 0.0002, respectively) (Figure 4C; Figure 4D).

### 3.3. Lack of Correlation Between Enzyme Expression and Neurosteroid Concentration

Next, we assessed whether the expression levels of CYP11A1 and CYP17A1 in the endometriosis lesion correlates with the concentrations in the peritoneal fluid of the respective neurosteroids they produce (i.e., PS and DHEAS). Simple linear regression did not reveal a significant correlation between the level of CYP11A1 mRNA expression and PS levels in the peritoneal fluid (R^2^ = 0.0017, F = 0.1226, *p* = 0.7272, Y = −0.0009793*X + 0.04541) nor between the level of CYP17A1 mRNA expression and DHEAS levels in the peritoneal fluid (R^2^ = 0.008805, F = 0.6396, *p* = 0.4265, Y = −0.09867*X + 3.894) (Figure 5). Within-group association also did not show any correlations.

### 3.4. Inflammatory Markers and Innervation

The literature shows that TRPM3 expression is upregulated in neurons that innervate inflamed tissue [21]. Thus, immunohistochemical (IHC) stains were performed to investigate the levels of inflammation in the deep endometriosis lesions between the different groups (Appendix A). To this aim, CD68^+^, a common monocyte marker, and Masson trichrome stain, indicating fibrosis, were used. The percentage of positive cells or area was investigated in a radius of 200 µm from the endometrial glandular foci (Figure 6A,B). Analysis of this small exploratory dataset showed no statistical significance towards increased numbers of CD68^+^ cells (one-way ANOVA with Tukey’s multiple comparisons test, 1.1 ± 0.5% vs. 0.9 ± 0.9% vs. 1.8 ± 1.3%) and collagen-rich area (one-way ANOVA with Tukey’s multiple comparisons test, 45.8 ± 4.5% vs. 52.4 ± 1.9% vs. 53.1 ± 7.1%), when comparing endometriosis patients without pain symptoms (Group 2) versus endometriosis patients with only DYSM (Group 3) and endometriosis patients with both DYSM and CPP (Group 4). To elucidate whether patients with higher pain scores have increased innervation, the percentage of positive area for nerve fibres in the ectopic lesions was compared between groups using an immunohistological staining for PGP9.5. The analysis of this dataset clearly identified nerve fibres in all the endometriotic lesions. However, the percentage of nerve-fibre-positive area per lesion was not significantly different between the three different endometriosis groups of no pain, DYSM, and DYSM + CPP (one-way ANOVA with Tukey’s multiple comparisons test, 0.4 ± 0.3% vs. 0.4 ± 0.2% vs. 0.3 ± 0.2%, respectively) (Figure 6C). Moreover, a previously described association between the number of macrophages and nerve fibres was also not detected in our study [42] (Appendix A).

## 4. Discussion

Endometriosis is a disease associated with life-long symptoms and high financial burden to patients and society [43]. Unfortunately, current treatment strategies remain inadequate, as approximately 50% of women with endometriosis experience symptom recurrence 5 years after receiving treatment [7]. Therefore, understanding and elucidating the association between disease elements and pain symptoms can help to achieve new and better therapeutic interventions. The ExPAND study included a prospective investigation into the association between endometriosis-associated pain symptoms and neurosteroids.

Interestingly, our results showed that mRNA expression of CYP17A1 is significantly upregulated in eutopic endometrial biopsies from endometriosis patients with severe pain symptoms when compared to healthy controls. This is in contrast with the literature, where it is reported that CYP17A1 is equally expressed in endometriosis patients compared to controls [12,44,45], however, there is no data available regarding patients’ pain symptoms. The increased CYP17A1 expression levels might result in higher intracellular DHEA levels. As DHEA is a weak agonist of TRPM3, elevated levels in pain patients could potentially induce channel activation. Although the endometrium is poorly innervated by sensory neurons [46], TRPM3 expression is reported in whole-tissue biopsies from both controls and endometriosis patients [36,47]. Moreover, in endometriosis and other painful gynaecologic disorders like adenomyosis and uterine fibroids, an increase in the number of nerve fibres can be observed [46,48]. As endometrial CYP17A1 levels are increased in patients with severe dysmenorrhea, we could hypothesise that DHEA and TRPM3 might play a key role in the pain perception during painful menstruation. These differences in CYP17A1 mRNA expression were not observed in ectopic endometriosis lesions. However, healthy control tissue was not available to make a similar statistic comparison, and endometriosis lesions show more cell heterogeneity compared to endometrium biopsies. For all other investigated genes (CYP11A1, SULT1E1, SULT2A1, and SULT2B1), no significant differences were found between the different groups in neither EB nor EL. Given the power of our study, we can conclude that there is no association between their expression levels and the DYSM and NCPP symptoms of patients with deep endometriosis lesions.

Using LC-MS/MS, concentrations of PS and DHEAS were measured in the peritoneal fluid of all groups. However, no statistical differences were observed for any of the neurosteroid TRPM3 agonists. Nevertheless, our data are still relevant for pain signalling, since earlier work showed that, at body core temperature (37 °C), both PS and DHEAS can activate TRPM3 in an overexpression system of HEK293 cells in the low micromolar range [27]. To confirm our hypothesis, calcium microfluorimetric experiments were performed on human stem-cell-derived sensory neurons. These data confirm the agonistic potential of DHEAS on endogenously expressed neuronal human TRPM3. Of note, the peritoneal concentrations of both neurosteroids in our study are within the range of previously reported plasma levels for PS and DHEAS [49,50]. Potentially, the levels of PS and DHEAS could even be increased locally at the site of the deep endometriosis lesion.

Even though our results showed that the levels of TRPM3 agonists in the peritoneal fluid are similar between all patient groups, the expression of TRPM3 at the innervating nerve ending might differ, as research has shown that TRPM3 expression is upregulated in dorsal root ganglions that innervate inflammatory tissue [21,24]. To further investigate this hypothesis, IHC stains for inflammatory markers were performed on an exploratory dataset. No differences were noted regarding CD68-positive cells or the size of collagen-rich area in deep endometriosis lesions between different pain subgroups. Finally, the literature describes that the density of nerve fibres is correlated with pain severity in endometriosis [6]. However, our study did not detect differences between Groups 2, 3, and 4 regarding nerve area based on PGP9.5 staining. Note that this IHC study was performed on a limited number of patient samples (*n* = 5) and therefore one must be careful in drawing final conclusions.

It is important to note that, in addition to TRPM3, other receptors may also contribute to endometriosis-associated pain, including members of the tropomyosin receptor kinase family (TrkA, TrkB, and TrkC) as well as the p75 neurotrophin receptor (p75NTR) [51].

We want to acknowledge some of the limitations of this study. First, the VAS pain score should reflect the DYSM and NCPP at the time of surgery in medication-free circumstances, but due to care standards and the WERF-EPHect design, questions regarding lifetime scores were used. Second, advanced methods like single-cell RNA sequencing and intratissue steroid profiling would be preferable, but no additional samples are available. Third, mRNA data offer limited insight into enzyme activity and long-term storage at −80 °C could influence the expression patterns [52]. Unfortunately, the expression levels of TRPM3 cannot be measured directly as selective antibodies for IHC are not available and functional TRPM3 measurements on human samples are currently not possible. In addition, other neurosteroids like progesterone and oestrogen, which play prominent roles in the pathogenesis, should also have been investigated. Finally, the menstrual cycle was not taken into account and only deep endometriosis lesions were investigated, reducing variability but limiting this study’s conclusions.

## 5. Conclusions

To conclude, the ExPAND study analysed the effect of neurosteroids on both mRNA and expression level in endometrial biopsies and deep endometriosis lesions, relating to patients’ pelvic pain symptoms. To this end, four distinct patient populations were delineated based on pain symptoms, using the WERF-EPHect questionnaire, thereby generating a unique patient dataset. Between these groups, we were able to observe a significant upregulation of CYP17A1 mRNA expression in endometrial biopsies of endometriosis patients with severe pain symptoms compared to healthy controls. However, there is no association between CYP11A1, SULT1E1, SULT2A1, and SULT2B1 expressions in deep lesions and endometriosis-associated pain. In addition, DHEAS and PS were measured in the peritoneal fluid, which were detected at concentrations that could have an impact on endometriosis-associated pain. Specifically, we showed for the first time that the neurosteroid DHEAS can activate human nociceptors via TRPM3 activation. Additional expression studies are required to further pinpoint the potential role of TRPM3 and neurosteroids in endometriosis-associated pain and to explore the downstream signaling cascade.

## Figures and Tables

**Figure 1 biomolecules-15-01352-f001:**
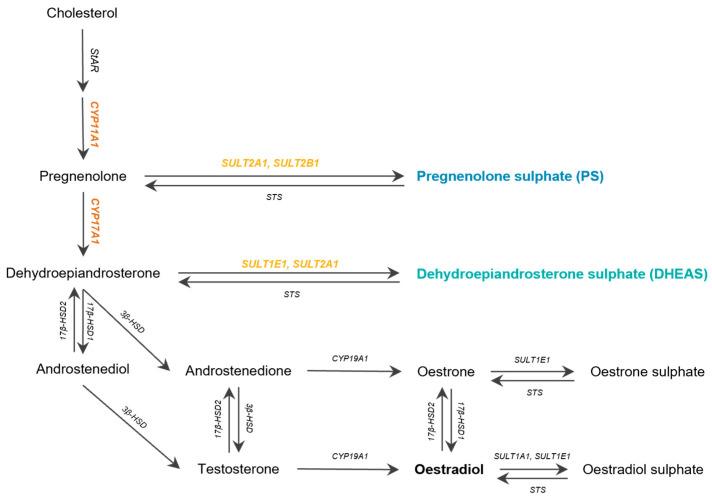
Overview of the steroidogenic enzymes assessed in the ExPAND study. CYP, Cytochrome P450 family; SULT, sulphotransferase; STS, steroid sulphatase.

**Figure 2 biomolecules-15-01352-f002:**
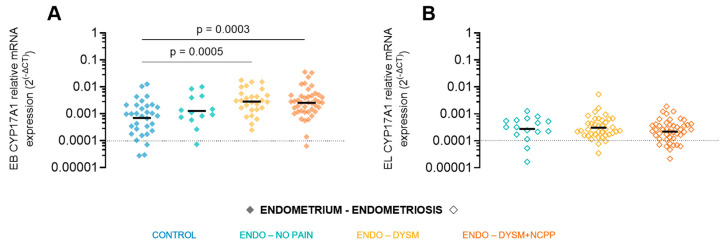
CYP17A1 mRNA expression levels of steroidogenic enzymes in eutopic endometrium of control and endometriosis patients. (**A**,**B**) mRNA expression levels of CYP17A1 in whole endometrium (closed symbols) and deep endometriosis (open symbols) lesions, respectively. These messenger RNA levels were quantified to the geometric mean of housekeeping genes GAPDH, HPRT1, and TBP. cDNA was synthesised from endometrial biopsies, sampled from the control group (Group 1, blue), endometriosis patients without pain symptoms (Group 2, green), endometriosis patients with only DYSM (Group 3, yellow), and endometriosis patients with both DYSM and NCPP (Group 4, red). Statistics: No outliers were detected using the ROUT method. One-way ANOVA with Dunnett’s correction was used for panel A: R^2^ = 0.1765; Group 1 vs. Group 3: adjusted *p* = 0.0002, 95% CI [0.8460–3.215]; Group 1 vs. Group 4: adjusted *p* = 0.0002, 95% CI [0.8085–2.935]. Kruskal–Wallis test with Dunn’s correction was used for panel B. ENDO, endometriosis; DYSM, dysmenorrhea; NCPP, non-cyclic pelvic pain; EB, endometrium biopsy; EL, endometriosis lesion.

**Figure 3 biomolecules-15-01352-f003:**
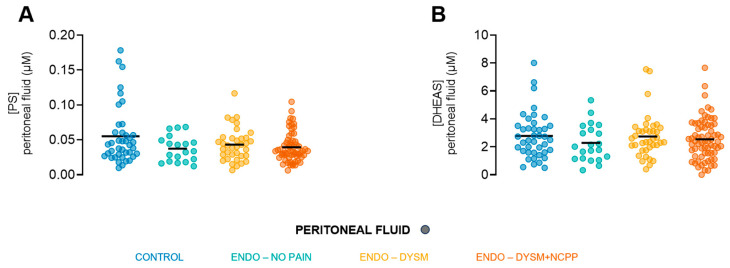
Concentrations of PS and DHEAS in peritoneal fluid of control and endometriosis patients. (**A**) Concentration of PS in the peritoneal fluid, sampled from the control group (Group 1), the endometriosis patients without pain symptoms (Group 2), endometriosis patients with only DYSM (Group 3), and endometriosis patients with both DYSM and NCPP (Group 4). (**B**) Concentration of DHEAS in the peritoneal fluid, sampled from the control group (Group 1), the endometriosis patients without pain symptoms (Group 2), endometriosis patients with only DYSM (Group 3), and endometriosis patients with both DYSM and NCPP (Group 4). Statistics: Outliers were detected and removed using the ROUT method. Kruskal–Wallis test with Dunn’s correction was used for panels A and B. ENDO, endometriosis; DYSM, dysmenorrhea; NCPP, non-cyclic pelvic pain; PS, pregnenolone sulphate; DHEAS, dehydroepiandrosterone.

**Figure 4 biomolecules-15-01352-f004:**
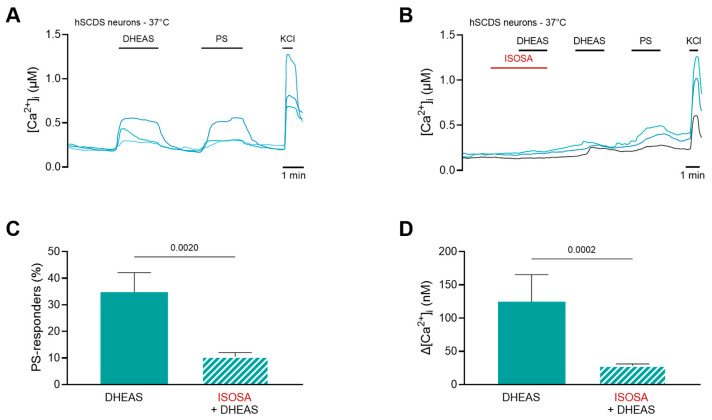
Human stem-cell-derived sensory (hSCDS) neurons showed responses towards DHEAS which could be blocked by TRPM3 inhibitor isosakuranetin. (**A**) Example traces of intracellular Ca^2+^ concentrations ([Ca^2+^]_i_) of hSCDS neurons upon application of 100 µM DHEAS followed by 40 µM PS and 50 mM KCl at 37 °C. (**B**) Representative traces of [Ca^2+^]_i_ of hSCDS neurons upon co-application of 5 µM isosakuranetin (ISOSA) with 100 µM DHEAS, followed by another application of 100 µM DHEAS, 40 µM PS, and 50 mM KCl at 37 °C. (**C**) Percentage (%) of DHEAS and DHEAS + ISOSA responders within the PS-responding neuronal cell population. (**D**) Intracellular Ca^2+^ influx, Δ[Ca^2+^]_i,_ upon application of 100 µM DHEAS or 5 µM ISOSA + 100 µM DHEAS within the PS-responding neuronal cell population. Mann–Whitney test was used for panel C, *p* = 0.002, 95% CI [7.281–43.21]. Mann–Whitney test was used for panel D, *p* = 0.0002, 95% CI [14.12–88.75]. N = 2219 cells over four independent differentiations. N_ISOSA_ = 1593 over three independent differentiations.

**Figure 5 biomolecules-15-01352-f005:**
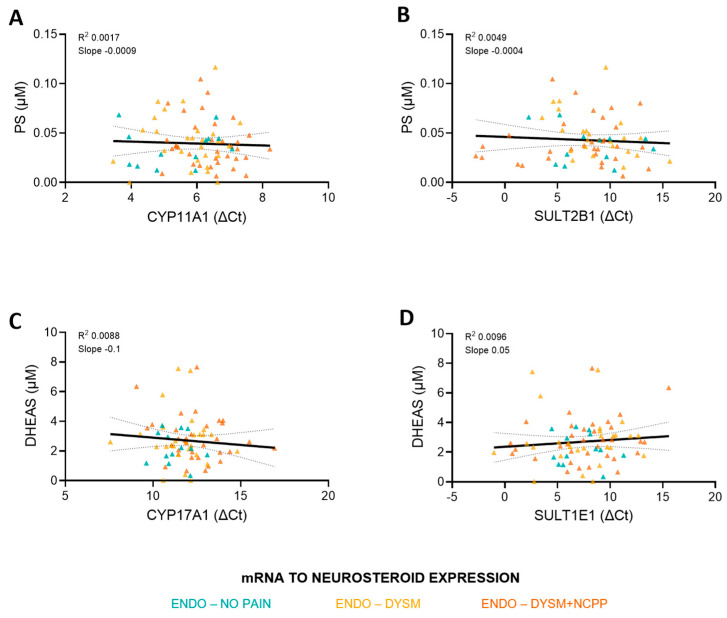
Simple linear regression in ectopic tissue expression and neurosteroid concentrations. (**A**,**B**) Simple linear regression correlation plot of CYP11A1 or SULT2B1 expression and PS concentration, respectively, in paired ectopic tissue and peritoneal fluid. Group 2 (green), Group 3 (orange), and Group 4 (red). (**C**,**D**) Simple linear regression correlation plot of CYP17A1 or SULT1E1 expression and DHEAS concentration in paired ectopic tissue and peritoneal fluid. Group 2 (green), Group 3 (orange), and Group 4 (red). ENDO, endometriosis; DYSM, dysmenorrhea; NCPP, non-cyclic pelvic pain.

**Figure 6 biomolecules-15-01352-f006:**
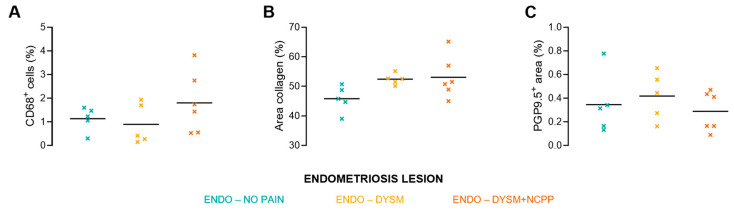
Inflammatory markers and nerve density in endometriosis lesions. (**A**) Percentage (%) of CD68-positive cells in deep endometriosis lesions from Groups 2, 3, and 4. (**B**) Percentage of collagen-rich area in deep endometriosis lesions from Groups 2, 3, and 4. (**C**) Percentage of nerve fibre area (PGP9.5) in deep endometriosis lesions from Groups 2, 3, and 4. One-way ANOVA with Tukey’s multiple comparisons test was used. Group 2 (green), Group 3 (orange), and Group 4 (red). ENDO, endometriosis; DYSM, dysmenorrhea; NCPP, non-cyclic pelvic pain.

**Table 1 biomolecules-15-01352-t001:** Patient demographics.

	Group 1 CONTROL		Group 2 ENDO NO PAIN		Group 3 ENDO DYSM		Group 4 ENDO DYSM + NCPP	
TOTAL N = 200	*n* = 44		*n* = 24		*n* = 54		*n* = 78	
**DYSMENORRHEA**								
NRS score (1–10)	ND		ND		8.5 ± 1.0		8.9 ± 0.9	
Pain medication								
None	ND		ND		5.6	(3)	3.8	(3)
Yes, NSAIDs	ND		ND		74.1	(40)	83.3	(65)
Yes, (weak) opioids	ND		ND		24.1	(13)	30.8	(24)
Yes, hormonal medication	ND		ND		24.1	(13)	38.5	(30)
**NON-CYCLIC PELVIC PAIN**								
NRS score (1–10)	ND		ND		0.4 ± 1.0		8.2 ± 1.4	
Pain medication								
None	ND		ND		98.1	(53)	17.9	(14)
Yes, NSAIDs	ND		ND		0	(0)	66.7	(52)
Yes, (weak) opioids	ND		ND		1.9	(1)	35.9	(28)
Yes, hormonal medication	ND		ND		0	(0)	20.5	(16)
**OTHER ENDOMETRIOSIS-ASSOCIATED PAIN SYMPTOMS**								
Dyspareunia	ND		12.5	(3)	61.1	(33)	75.6	(59)
Dyschezia	ND		12.5	(3)	68.5	(37)	70.5	(55)
Dysuria	ND		4.2	(1)	27.8	(15)	46.2	(36)
**rAFS POINTS**								
	ND		45.0 ± 28.6		58.0 ± 34.4		53.9 ± 34.5	
rAFS Stage 1			0	(0)	0	(0)	0	(0)
rAFS Stage 2			16.6	(4)	18.5	(10)	21.8	(17)
rAFS Stage 3			29.2	(7)	9.3	(5)	14.1	(11)
rAFS Stage 4			54.2	(13)	72.2	(39)	64.1	(50)
**#ENZIAN SCORE**								
Compartment A (vagina, rectovaginal space)	ND		1.0 ± 1.1		1.9 ± 1.0		1.8 ± 1.2	
Score 0			45.8	(11)	14.8	(8)	23.1	(18)
Score 1			20.8	(5)	13.0	(7)	12.8	(10)
Score 2			20.8	(5)	38.9	(21)	28.2	(22)
Score 3			12.6	(3)	33.3	(18)	35.9	(28)
Compartment B left (USL, cardinal ligaments, and pelvic sidewall)	ND		0.7 ± 0.8		1.3 ± 1.0		1.1 ± 1.0	
Score 0			50.0	(12)	29.6	(16)	35.9	(28)
Score 1			29.2	(7)	20.4	(11)	18.0	(14)
Score 2			20.8	(5)	42.6	(23)	42.3	(33)
Score 3			0	(0)	7.4	(4)	3.8	(3)
Compartment B right (USL, cardinal ligaments, and pelvic sidewall)	ND		1.0 ± 0.8		0.7 ± 0.8		1.0 ± 1.0	
Score 0			37.5	(9)	57.4	(31)	46.1	(36)
Score 1			29.2	(7)	22.2	(12)	16.7	(13)
Score 2			33.3	(8)	18.5	(10)	32.1	(25)
Score 3			0	(0)	1.9	(1)	5.1	(4)
Compartment C (rectum)	ND		0.9 ± 1.2		1.3 ± 1.3		1.4 ± 1.3	
Score 0			58.3	(14)	46.3	(25)	41.0	(32)
Score 1			4.2	(1)	5.6	(3)	10.3	(8)
Score 2			25.0	(6)	20.4	(11)	19.2	(15)
Score 3			12.5	(3)	27.8	(15)	29.5	(23)
**HORMONAL MEDICATION**								
None	77.2	(34)	4.2	(1)	0	(0)	1.3	(1)
GnRH agonist	4.5	(2)	33.3	(8)	63.0	(34)	64.1	(50)
Combination OAC	9.5	(4)	54.2	(13)	20.3	(11)	21.8	(17)
Progesterone only OAC	4.5	(2)	8.3	(2)	16.7	(9)	12.8	(10)
Aromatase inhibitor	4.5	(2)	0	(0)	0	(0)	0	(0)
**AGE**								
	37.0 ± 6.2		34.2 ± 3.3		31.2 ± 3.7		32.6 ± 5.5	
< 25 years	0.0	(0)	0	(0)	5.5	(3)	5.1	(4)
25–35 years	34.1	(15)	70.8	(17)	78.2	(42)	70.5	(55)
35 + years	65.9	(29)	29.2	(7)	16.3	(9)	24.4	(19)

The ExPAND study population consists of four distinct patient groups, i.e., control (1), deep endometriosis—no pain symptoms (2), deep endometriosis—only severe dysmenorrhea (3), deep endometriosis—severe dysmenorrhea and non-cyclic pelvic pain (4). Inclusion criteria were set to define the presence of severe dysmenorrhea and non-cyclic pelvic pain: DYSM NRS ≥ 7 and NCPP NRS ≥ 5. Patient demographics such as pain medication usage and age were also recorded. Regarding the presence of endometriosis, disease staging tools were consulted, i.e., rAFS score and #Enzian score [17,18]. DYSM, dysmenorrhea; NCPP, non-cyclic pelvic pain, NR, S numeric rating scale, ND, not determined; OAC, oral contraceptive. Mean ± SD, *%*, (n).

## Data Availability

The original contributions presented in this study are included in the article/Appendix A. Further inquiries can be directed to the corresponding author(s).

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
