# Peer review of "The ExPAND Study: A Prospective Association Study into Endometriosis-Associated Pain, Neurosteroid Synthesis, and TRPM3"

_biomolecules, 2025, doi:10.3390/biom15101352_

Round 1

Reviewer 1 Report

Comments and Suggestions for Authors

Dear Authors,

The article “The ExPAND study: a prospective association study between  neurosteroids, their expression and endometriosis-associated pain» describes the role of neurosteroids pregnenolone sulphate and dehydroepiandrosterone sulphate and TRPM3 in the endometriosis-associated pain.

There study is actual and interesting, since the role of neurosteroids in the treatment of neurological diseases and pain development is a modern task of pharmacology.

There are some comments below.

  1. Progesterone and allopregnanolone are well-known neurosteroid. P4 plays the central role in the pathogenesis of endometriosis. Why did you choose for the analysis DHEAS and PS, please, discuss.
  2. There are well known neurotrophins receptors which contribute to the pain such as TrkA, TrlB, Trkc, p75NTR. Please, discuss their role in the endometriosis-associated pain. My proposal is to include TRPM3 in the Title.
  3. The tissue samples were frozen for a long time. Please, explain in the Methods section how the storage could influence the expression patterns with the references to the known and established procedures which confirm the possibility of such a prospective analysis.
  4. There are ABB (Pen, Strep) and codes such as : 1 μM small molecule inhibitors, 1 μM LDN193189 and 10 μM SB-431542: inhibitors of what?
  5. Please, add the definition of N2 supplement, B27 supplement, medium supplemented with 5 μM CHIR99021, 5 μM DAPT, 5 μM SU5402
  6. There are mistakes: “Aetiology”
  7. It is desirable to add graphical abstract.

Reviewer 2 Report

Comments and Suggestions for Authors

This is a thoughtful and carefully conducted study that prospectively connects symptom-defined endometriosis groups with steroidogenic enzyme expression, peritoneal PS/DHEAS levels, and TRPM3-dependent neuronal responses. The clinical phenotyping (WERF-EPHect), prospective sampling of eutopic and ectopic tissues, and mechanistic validation using hSCD neurons are notable strengths. I consider this a solid paper with clear potential impact once the following minor issues are addressed.

  1. Report adjusted p-values alongside effect sizes and confidence intervals.
  2. The use of ROUT (Q = 1%) to remove outliers may influence findings in a heterogeneous clinical dataset. Please pre-specify outlier rules, include sensitivity analyses with and without outlier removal, or use rank-based tests/robust regression to minimize data exclusion.
  3. Clearly state covariates included in models (age, BMI, treatment type), particularly for ANCOVA or regression analyses.
  4. The IHC sample size appears small; “no difference” conclusions should be framed as exploratory.
  5. Provide a brief computational appendix (R scripts, key code snippets, and software version manifest).
  6. In Figure 5, the caption describes “simple linear regression correlation plots,” but the plots show Spearman r. This mismatch is confusing: Spearman is rank-based, while the regression line suggests a linear model. Please either report R² if using linear regression or remove the regression line if Spearman is used, and clarify in the figure legend and Methods.
  7. Consider exploring within-group associations separately, even if underpowered, to assess whether enzyme-steroid relationships are subgroup-driven or consistent across all groups.
  8. Consolidate key limitations, such as treatment and cycle-phase confounding, use of peritoneal rather than intralesional steroid levels, and limited IHC sample size, into a concise paragraph at the end of the Discussion. 

Round 2

Reviewer 1 Report

Comments and Suggestions for Authors

The article needs improvements.

First of all, the title of the article has no sense, since it can`t be “the expression of neurosteroids”.

The Figures do not contain all needed information. The reader must download supplementary materials to understand some ABB.

Table S3 does not contain information about CHIR99021, DAPT, SU5402, LDN193189 and SB-431542. It is still not explained.  Chemical pathway inhibitors have concrete indications. The name of this table: Supplementary Table S3. Overview of immunohistology stain protocols

Table S4 (line 321) is absent.

isosakuranetin (ISOSA) is firstly abbreviated after Fig.4., but mentioned before

The authors wrote they added GA. It is absent.
